# Microstructure and Morphology Control of Potassium Magnesium Titanates and Sodium Iron Titanates by Molten Salt Synthesis

**DOI:** 10.3390/ma12101577

**Published:** 2019-05-14

**Authors:** Haoran Zhang, Mengshuo Li, Ze Zhou, Liming Shen, Ningzhong Bao

**Affiliations:** State Key Laboratory of Material-Oriented Chemical Engineering, College of Chemical Engineering, Nanjing Tech University, Nanjing 210009, China; zhr111@njtech.edu.cn (H.Z.); lmshlove@njtech.edu.cn (M.L.); 1461749960@njtech.edu.cn (Z.Z.)

**Keywords:** titanates, morphology, molten salt, crystal growth, Formation

## Abstract

Titanates materials have attracted considerable interest due to their unusual functional and structural properties for many applications such as high-performance composites, devices, etc. Thus, the development of a large-scale synthesis method for preparing high-quality titanates at a low cost is desired. In this study, a series of quaternary titanates including K_0.8_Mg_0.4_Ti_1.6_O_4_, Na_0.9_Mg_0.45_Ti_1.55_O_4_, Na_0.75_Fe_0.75_Ti_0.25_O_2_, NaFeTiO_4_, and K_2.3_Fe_2.3_Ti_5.7_O_16_ are synthesized by a simple molten salt method using inexpensive salts of KCl and NaCl. The starting materials, intermediate products, final products, and their transformations were studied by using TG-DSC, XRD, SEM, and EDS. The results show that the grain size, morphology, and chemical composition of the synthesized quaternary titanates can be controlled simply by varying the experimental conditions. The molar ratio of mixed molten salts is critical to the morphology of products. When KCl:NaCl = 3:1, the morphology of K_0.8_Mg_0.4_Ti_1.6_O_4_ changes from platelet to board and then bar-like by increasing the molar ratio of molten salt (KCl–NaCl) to raw materials from 0.7 to 2.5. NaFeTiO_4_ needles and Na_0.75_Fe_0.75_Ti_0.25_O_2_ platelets are obtained when the molar ratio of molten salt (NaCl) to raw materials is 4. Pure phase of Na_0.9_Mg_0.45_Ti_1.55_O_4_ and K_2.3_Fe_2.3_Ti_5.7_O_16_ are also observed. The formation and growth mechanisms of both potassium magnesium titanates and sodium iron titanates are discussed based on the characterization results.

## 1. Introduction

The titanates is a group of inorganic compounds consisting of titanium, oxygen, and one or more other metallic elements. Dependent on the linkage method of the structure unit of TiO_6_ octahedra, titanates may exhibit cage, tunnel, and layered structures. The commonly employed titanates in industrial applications include CaTiO_3_, BaTiO_3_, SrTiO_3_, and M_2_O.nTiO_2_ (M = K/Na, n = 1~8), covering a wide range from the medical to electrical and automotive industries. As opposed to the ternary titanates with well-investigated properties and developed applications, quaternary titanates are still in the exploring stage, partly due to the large range of compositions and the complex structural deviation. Among all quaternary titanates, K_2_O-MgO-TiO_2_ and K_2_O/Na_2_O-Fe_2_O_3_-TiO_2_ are the most studied systems to investigate the reaction boundary and crystallographic structure. In order to represent both the stoichiometric and nonstoichiometric compositions, two forms of formula, A*_x_*(B*_x_*_/2_^II^Ti*_y_*_−*x*/2_)O_2*y*_ and A*_x_*(B*_x_*^III^Ti*_y_*_−*x*_)O_2*y*_ (*y* = 1, 2, 4, 8) with A = alkali metal ions and B = divalent (B^II^) or trivalent (B^III^) ions, are generally used.

The A*_x_*(B*_x_*_/2_^II^Ti_8−*x*/2_)O_16_ and A*_x_*(B*_x_*^III^Ti_8−*x*_)O_16_, with the *x* value in the range of 1.5 < *x* ≤ 2.0, have hollandite structure. Fujiki et al. [1] prepared hollandite K_1.6_Mg_0.8_Ti_7.2_O_16_ and K_1.6_Al_1.6_Ti_6.4_O_16_ crystals using the K_2_MoO_4_-MoO_3_ flux melting method. Endo et al. [2] obtained hollandite K_1.6_Fe_1.6_Ti_6.4_O_16_ in the shape of a needle and α-phase K_2.3_Fe_2.3_Ti_5.7_O_16_ in the shape of platelet using a similar synthesis method and observed the ionic conductivity. Park et al. [3] generated a composite of short fibers of hollandite K_2_MgTi_7_O_16_ and glass with enhanced bending strength. Chen et al. [4] synthesized hollandite K_1.54_Mg_0.77_Ti_7.33_O_16_ whiskers with high near-infrared reflectivity. Akieh et al. [5] prepared Na_2_Fe_2_Ti_6_O_16_ by solid-state method and investigated the ion exchange properties by removing ionic. Ni. Knyazev et al. [6] investigated the structure and thermal expansion of synthesized K_2_Fe_2_Ti_6_O_16_ with hollandite structure and Na_2_Fe_2_Ti_6_O_16_ with freudenbergite structure.

The research on A*_x_*(B*_x_*_/2_^II^Ti_4−*x*/2_)O_8_ and A*_x_*(B*_x_*^III^Ti_4−*x*_)O_8_ is relatively rare. Hou et al. [7] reported on the solid-state reaction synthesis of sodium titanate bronze-type NaFeTi_3_O_8_ as an anode material for sodium-ion batteries exhibiting a discharge capacity of 170.7 mA·h·g^−1^ at a current density of 20 mA·g^−1^. NaFeTiO_4_ and K_0.8_Mg_0.4_Ti_1.6_O_4_ are the two representative compounds for A*_x_*(B*_x_*_/2_^II^Ti_2−*x*/2_)O_4_ and A*_x_*(B*_x_*^III^Ti_2−*x*_)O_4_, respectively. NaFeTiO_4_ is a calcium-ferrite type octatitanate. Archaimbault et al. [8] found that Na_0.875_Fe_0.875_Ti_1.125_O_4_ is a unique composition. Sodium titanate bronze-type NaFeTi_3_O_8_ (*x* < 0.875) and calcium-ferrite type NaFeTiO_4_ (*x* > 0.875) were found on each side of the composition *x* = 0.875. Mumme et al. [9] prepared Na*_x_*Fe*_x_*Ti_2−*x*_O_4_ with a small structural variation within the range of 0.75 < *x* < 0.9. Kuhn et al. [10] conducted sodium extraction on Na_0.875_Fe_0.875_Ti_1.125_O_4_ and studied the conductivity. K_0.8_Mg_0.4_Ti_1.6_O_4_ has lepidocrocite-like structure and is used to produce friction materials. Tan et al. [11] synthesized K_0.8_Mg_0.4_Ti_1.6_O_4_ platy powders by a flux method to remove copper ions from water pollutants through ion-exchange adsorption. Liu et al. [12] prepared K_0.8_Mg_0.4_Ti_1.6_O_4_ platelets and porous ceramics to remove Ni ions from wastewater.

Both A*_x_*(B*_x_*_/2_^II^Ti_1−*x*/2_)O_2_ and A*_x_*(B*_x_*^III^Ti_1−*x*_)O_2_ have attracted increased interest due to the potential application of layered α-NaFeO_2_ structure for Na-ion cathode materials. Li et al. [13] prepared Na_1−*x*_Fe_1−*x*_Ti*_x_*O_2_ (0 ≤ *x* ≤ 0.28) with α-NaFeO_2_ structure and K_1−*x*_Fe_1−*x*_Ti*_x_*O_2_ (0 ≤ *x* < 0.20) with β-cristobalite structure. Fujishiro et al. [14] synthesized Na_0.4_M_0.2_Ti_0.8_O_2_ (M = Co, Ni, and Fe) with α-NaFeO_2_ structure and measured their thermoelectric properties. Thorne et al. [15] studied the structural stabilization of iron containing cathode materials by substituting some iron in α-NaFeO_2_ with titanium to produce Na*_x_*Fe*_x_*Ti_1−*x*_O_2_ (0.75 ≤ *x* ≤ 1.0).

The studies above mainly focus on the investigation of compositional and structural variations and the potential applications on specific compositions. Various compositions have been made by different kinds of synthesis methods, such as high-temperature calcination, molten salt synthesis, kneading-drying-calcination (KDC), etc. [1,2,3,4,5,6,7,8,9,10,11,12,13,14,15]. The subject of this study is to achieve a stable production of high-quality quaternary octatitanates with controllable morphology and narrow size distribution for potential applications in inorganic fiber-reinforced composites and sodium ion batteries. The molten salt method and low-cost raw materials have thus been exclusively used for future scalable industry production. Through the adjustment of the content (α) of molten salt in raw materials, the ratio (β) of KCl in KCl–NaCl molten salt, and the reaction temperature and time, we obtained pure phase of lepidocrocite-like K_0.8_Mg_0.4_Ti_1.6_O_4_ and Na_0.9_Mg_0.45_Ti_1.55_O_4_, calcium-ferrite type NaFeTiO_4_, Na_0.75_Fe_0.75_Ti_0.25_O_2_ with layered α-NaFeO_2_ structure, and α-phase K_2.3_Fe_2.3_Ti_5.7_O_16_. Three KMTO products, namely K_0.8_Mg_0.8_Ti_1.6_O_4_ platelets, K_0.8_Mg_0.8_Ti_1.6_O_4_ boards, and K_0.8_Mg_0.8_Ti_1.6_O_4_ bars, are obtained at three sets of optimum conditions (α = 0.7, 1.5, and 2.5; β = 0.75; T=1050 °C; t = 4 h). Two NFTO products, namely NaFeTiO_4_ needles and Na_0.75_Fe_0.75_Ti_0.25_O_2_ platelets, are obtained when T = 900 and 1000 °C; α = 4; β = 0; and t = 4. The products and their intermediate products are characterized by scanning electron microscopy, X-ray diffraction, and thermogravimetric analysis for a better understanding of their formation and growth processes. The current synthesis procedure can be scaled for controllable production of these types of titanates.

## 2. Materials and Methods

### 2.1. Reagents and Materials

Titanium dioxide (TiO_2_), iron oxide (Fe_2_O_3_), and magnesium carbonate basic pentahydrate (4MgCO_3_·Mg(OH)_2_·5H_2_O) were purchased from Sinopharm Chemical Reagent Co., Ltd. (Shanghai, China). Potassium carbonate (K_2_CO_3_), sodium carbonate (Na_2_CO_3_), sodium chloride (NaCl), and potassium chloride (KCl) were purchased from Shanghai Lingfeng Chemical Reagent Co., Ltd., China. Ilmenite (FeTiO_3_) was purchased from Jiangsu Taibai Group Co., Ltd. (Zhenjiang, China). The alumina crucibles (50 mL) used for calcination were purchased from Nanjing Wanqing Chemical Glass Instrument Co., Ltd. (Nanjing, China).

### 2.2. Adjustment of Molten Salt Content

The molten salt content α is defined as the molar ratio of molten salt to raw materials, namely α = n_molten salt_/n_raw materials_. The raw materials include the molten salt and the starting materials (see 2.3 and 2.4). The molten salt ratio β is defined as the molar ratio of KCl to KCl–NaCl, namely β = n_KCl_/n_mixture of NaCl-KCl_.

### 2.3. Preparation of Potassium Magnesium Titanates (KMTO)

Potassium magnesium titanates were produced by the molten salt synthesis method. The starting materials of TiO_2_, K_2_CO_3_, and 4MgCO_3_·Mg(OH)_2_·5H_2_O (Ti: K: Mg = 4:2:1) were mixed with KCl–NaCl molten salt (β = 0, 0.3, 0.45, 0.75, 0.85, and 1) at a certain molar ratio (α = 0, 0.5, 0.7, 1.5, 2.0, 2.5, and 6.0). The mixture was calcined at 750, 850, 950, and 1050 °C for 2, 4, or 6 h for the procedure optimization. After cooling to room temperature, the product was washed with distilled water to remove salt residue and was then dried at 100 °C for 10 h. Three kinds of K_0.8_Mg_0.4_Ti_1.6_O_4_ (KMTO) products, namely KMTO platelets, boards, and bars obtained at three sets of optimum conditions (α = 0.7, 1.5, and 2.5; β = 0.75; T = 1050 °C, t = 4 h), were discussed in detail.

### 2.4. Preparation of Sodium Iron Titanates (NFTO)

Sodium iron titanates were produced by the same procedure as described in the preparation of KMTO. The starting materials of Na_2_CO_3_ and FeTiO_3_, with or without Fe_2_O_3_, were mixed with KCl–NaCl molten salt (β = 0, 0.25, 0.5, 0.75, and 1) at a certain molar ratio (α = 2, 4, and 6). The mixture was calcined at 600, 700, 800, 900, and 1000 °C for 2, 4, or 6 h for the procedure optimization. Two kinds of NFTO products, NaFeTiO_4_ needles and Na_0.75_Fe_0.75_Ti_0.25_O_2_ platelets, were obtained at two sets of optimum conditions. Without Fe_2_O_3_ as the Fe source, NaFeTiO_4_ needles were prepared at the condition of Na:Fe:Ti = 1.3:1:1, α = 4, β = 0, T = 900 °C, and t = 4 h. With the presence of Fe_2_O_3_, Na_0.75_Fe_0.75_Ti_0.25_O_2_ platelets were obtained during the condition of Na:Fe:Ti = 3.3:2.2:1, α = 4, β = 0, T = 1000 °C, and t = 4 h.

### 2.5. Characterizations

The crystalline phase of samples were examined with X-ray diffraction (XRD) by using a D8-Advance, Bruker AXS diffractometer (Cu-Kα radiation, λ = 1.5418 Å) in the continuous scan mode over 5–70° (2θ) with a scan rate of 0.3°/s, operating at 40 kV and 40 mA. The morphology and microstructure of samples were characterized by field-emission scanning electron microscopy (FESEM, HITACHI S−4800, Hitachi, Tokyo, Japan) equipped with energy-dispersive X-ray spectroscopy (EDS). Thermogravimetric analysis (TGA) was performed on a NETZSCH 449 STA thermogravimetric analyzer (Netzsch, Sabre, Germany). The samples were heated in N_2_ atmosphere from 30 to 1100 °C at a heating rate of 10 °C·min^−1^.

## 3. Results and Discussion

### 3.1. Synthesis of KMTO with Different Morphologies

Figure 1 and Figure 2 show the SEM images and XRD patterns of the KMTO samples prepared at different molar ratios of the molten salt KCl–NaCl to the raw materials (α = 0.7, 1.5, and 2.5) after calcination at 1050 °C for 4 h. The morphologies of KMTO products are shaped like platelets (α = 0.7, Figure 1a1,a2), boards (α = 1.5, Figure 1b1,b2), and bars (α = 2.5, Figure 1c1,c2), respectively. As shown in Figure 2, the major phase of all the three differently shaped products exhibit the XRD patterns belonging to K_0.8_M_g0.4_Ti_1.6_O_4_ (PDF#35-0046). A very small amount of impurity peaks of hydrated potassium tianium hydrogen oxide hydrate (K_0.5_H_1.5_Ti_4_O_9_·0.6H_2_O) is also observable in KMTO boards and bars (Figure 2c,d). This impurity phase may be due to the dissolution of K^+^ and Mg^+^ when the products were washed by DI water. The relative peak intensity of three samples differs from that of the standard XRD pattern and the product synthesized without the presence of molten salts (α = 0, Figure 2a). The deviation of the peak intensity is caused by the preferential growth of samples. The elemental analysis from EDS is consistent with XRD, see Appendix A. Appendix A show the products prepared at different reaction temperatures (750, 850, and 950 °C) and different reaction durations (2, 4, and 6 h). The pure phase KMTO obtained at 1050 °C for 4 h have obviously better morphology control.

The growth mechanism of the KMTO platelets, boards, and bars are proposed as depicted in Figure 3, based on the melting point of the molten salt (675 °C for β = 0.75), calculations on the thermodynamics of the reactions between sodium and potassium cations, and analyses using SEM (Appendix A), XRD (Appendix A), and TG–DSC (Appendix A) on the intermediate products during the entire heating process. As the calcination temperature increases, 4MgCO_3_·Mg(OH)_2_·5H_2_O first loses hydration water and then decomposes to MgO and CO_2_. Upon the dissolution of K_2_CO_3_ and MgO in the NaCl-KCl molten salt, K^+^ and Mg^2+^ ions diffuse at different rates in the liquid phase, approaching the dispersed TiO_2_ particles. When α = 0.7, KMTO particles are directly formed and then gradually evolve to crystalline platelets as the temperature reaches 1050 °C. When α ≥ 1.5, with abundant Na^+^ in the system, low-melting intermediate phase Na_8_Ti_5_O_14_ (melting point 965~985 °C) is formed first and then it interacts with Mg^2+^ and K^+^ in the melt to form more stable NMTO bars (melting point 1100 °C). Based on the thermodynamic calculation, the ion exchange from Na^+^ to K^+^ will spontaneously occur when the system temperature is above 675 °C. So as the temperature continues to increase, Na^+^ in NMTO exchanges with K^+^ from the molten salt, resulting in a more stable high melting KMTO phase (melting point 1300 °C) which retains a long strip shape. The morphology and crystalline phase of the intermediate NMTO phase were confirmed by characterizing the samples rapidly annealed at 750, 850, and 950 °C by using SEM and XRD (Appendix A).

This kind of morphology control cannot be obtained by using KCl (β = 1) or NaCl (β = 0) alone as the molten salt. Figure 4 and Figure 5 show the XRD patterns and SEM images of the samples prepared at different mole ratios of the molten salt to the raw materials (α = 0.5, 2, and 6) after calcination at 1050 °C for 4 h. When using KCl as the molten salt (β = 1), all three samples are pure KMTO phase (Figure 4a1–a3). As the molar ratio α increases from 0.5 to 2 and 6, the relative peak intensity changes. However, all three samples have the platelet morphology of several micrometers (Figure 5a1–a3), indicating that single KCl molten salt cannot cause the platelet-board-bar morphology evolution as KCl–NaCl. When using single NaCl as molten salt (β = 0), a new phase of sodium magnesium titanate (Na_0.9_Mg_0.45_Ti_1.55_O_4_) appears and the product becomes slender as the amount of NaCl in the molten salt increases. At α = 0.5, the product is a mixture of KMTO particles and NMTO bars (Figure 4b1 and Figure 5b1). When α increases to 2, the KMTO phase disappears and the product becomes pure NMTO rods (Figure 4b2 and Figure 5b2), indicating that NaCl in the molten salt provided Na^+^ to participate in the crystal growth reaction. When α increases to six, the product is long NMTO whiskers (Figure 4b3 and Figure 5b3). Although the equilibrium constant for NaCl(*l*) + K^+^(*s*) → Na^+^(*s*) + KCl(*l*) is 1 [16], the K^+^ ions in the layer structured KMTO can still be displaced by Na^+^ via ion exchange when the concentration of surrounding Na^+^ ions is sufficiently large. The above experimental results indicate that the molten salt does not only provide a liquid phase environment for reactions, however the cations may also participate in reactions and strongly affect the growth of crystalline products.

### 3.2. Synthesis of NFTO with Different Morphologies

Figure 6 and Figure 7 show the XRD pattern and SEM images of the NFTO samples synthesized while using NaCl as the molten salt. At the condition of Na:Fe:Ti = 1.3:1:1 and Na:Fe:Ti = 3.3:2.2:1, the XRD patterns of the products can be assigned to NaFeTiO_4_ (PDF#33-1255, Figure 6a) and Na_0.75_Fe_0.75_Ti_0.25_O_2_ (PDF#25-0877, Figure 6b), respectively. None of the noticeable peaks belong to the unreacted reactants (Na_2_CO_3_) or intermediate phase (Fe_2_O_3_), indicating that the starting materials have been completely transformed to products at appropriate annealing temperatures (NaFeTiO_4_, 900 °C; Na_0.75_Fe_0.75_Ti_0.25_O_2_, 1000 °C). The chemical composition of products is also confirmed by EDS analyses (Appendix A). As shown in Figure 7, NaFeTiO_4_ is in the shape of needles with the length of 20–50 μm and diameter of 0.5–2 μm, while Na_0.75_Fe_0.75_Ti_0.25_O_2_ has the platelet shape with the size range of 5–20 μm. The products show the best morphology when the molar ratio of NaCl to the raw materials is α = 4 (Figure 8). The influence of the ratio of reactants on the product was also investigated and the condition of Na:Fe:Ti = 3.3:2.2:1 shows the pure phase Na_0.75_Fe_0.75_Ti_0.25_O_2_ with relatively uniform morphology (Appendix A). Appendix A show the products prepared at different reaction temperatures (600, 700, 800, 900, and 1000 °C) and different reaction durations (2, 4, and 6 h). NaFeTiO_4_ needles with best morphology were obtained at 900 °C for 4 h and Na_0.75_Fe_0.75_Ti_0.25_O_2_ platelets were obtained at 1000 °C for 4 h.

The growth mechanism of NFTO is proposed as depicted in Figure 9, based on the analyses using SEM (Appendix A), XRD (Appendix A), and TG-DSC (Appendix A). As the calcination temperature increases, free water in the raw materials gets released and Na_2_CO_3_ decomposes to Na_2_O and CO_2_ below 700 °C. After FeTiO_3_ is completely converted to Fe_2_O_3_ and Fe_2_Ti_3_O_9_ around 620 °C, the reaction system changes from Na_2_CO_3_-FeTiO_3_ to Na_2_O-Fe_2_O_3_-Fe_2_Ti_3_O_9_. The NaFeTiO_4_ phase starts to appear after 700 °C. The Fe_2_O_3_ and Fe_2_Ti_3_O_9_ were completely consumed at 900 °C. The product obtained at 900 °C exhibits the best needle-like morphology and a relatively narrow size dispersion. The average diameter and length of the as-prepared NaFeTiO_4_ needles are in the range of 0.5–2 μm and 20–50 μm, respectively. At 1000 °C, part of NaFeTiO_4_ starts to break into small pieces. Fe_2_O_3_ was added in the starting materials to increase the Fe content. NaFeTiO_4_ with low crystallinity forms at 700 °C. With sufficient Fe source, Na_0.75_Fe_0.75_Ti_0.25_O_2_ platelets start to appear at 800 °C. Thus, Na_0.75_Fe_0.75_Ti_0.25_O_2_ platelets and a small amount of NaFeTiO_4_ rods are both present in products from 800–900 °C. At 1000 °C, NaFeTiO_4_ phase is disappeared and pure phase Na_0.75_Fe_0.75_Ti_0.25_O_2_ platelets are obtained.

To investigate the influence of molten salt type on the growth of NFTO, KCl-NaCl composite molten salt with different ratios were used for reactions. Figure 10 and Figure 11 show the XRD patterns and SEM images while using KCl–NaCl as the composite molten salt and α is fixed at 4. When β is below 0.5, only the NaFeTiO_4_ phase is detectable. At β = 0.25, the product has both rod-like and platelet shapes. At β = 0.5, the rods are apparently larger in size, accompanied with randomly shaped particles. While β is 0.75, the product is a mixture of NaFeTiO_4_ and K_2.3_Fe_2.3_Ti_5.7_O_16_ and the product contains both big rods and random particles. When the KCl content reaches 100% (β = 1), pure phase K_2.3_Fe_2.3_Ti_5.7_O_16_ is observed. The morphology changes to a mixture of large plates and small particles. Hence, the results indicate that KCl in the molten salt can participate in the crystal growth of NFTO and should be avoided for obtaining pure phase NFTO.

## 4. Conclusions

We have prepared a series of quaternary KMTO and NFTO titanates by using the molten salt synthesis method. The molar ratio of molten salt is critical to the composition and morphology of products. The KCl–NaCl molten salt is preferred for growing KMTO. When α = 0.7, 1.5, and 2.5 (β = 0.75), K_0.8_Mg_0.8_Ti_1.6_O_4_ platelets, K_0.8_Mg_0.8_Ti_1.6_O_4_ boards, and K_0.8_Mg_0.8_Ti_1.6_O_4_ bars were obtained, respectively, after calcination at 1050 °C for 4 h. NMTO was prepared by using NaCl alone as the molten salt. When α = 4 and β = 0, NaFeTiO_4_ needles and Na_0.75_Fe_0.75_Ti_0.25_O_2_ platelets were obtained. The calcination temperatures were 900 and 1000 °C, respectively. The well-controlled morphology is useful for practical applications.

## Figures and Tables

**Figure 1 materials-12-01577-f001:**
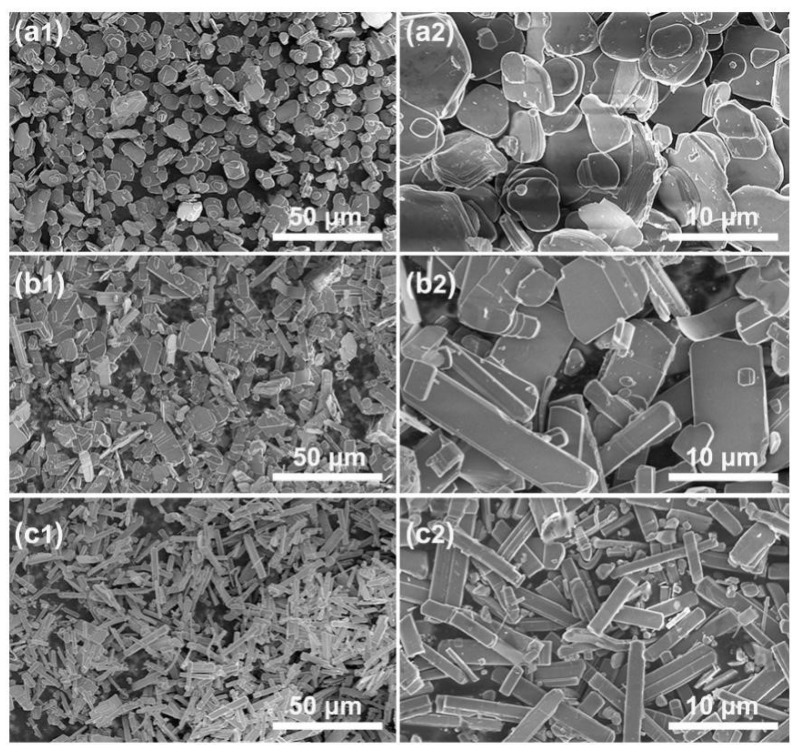
SEM images of three differently shaped KMTO products prepared at 1050 °C for 4 h with different molar ratios of the molten salt to the raw materials: (**a1**,**a2**) platelets, α = 0.7, β = 0.75; (**b1**,**b2**) boards, α = 1.5, β = 0.75; and (**c1**,**c2**) bars α = 2.5, β = 0.75.

**Figure 2 materials-12-01577-f002:**
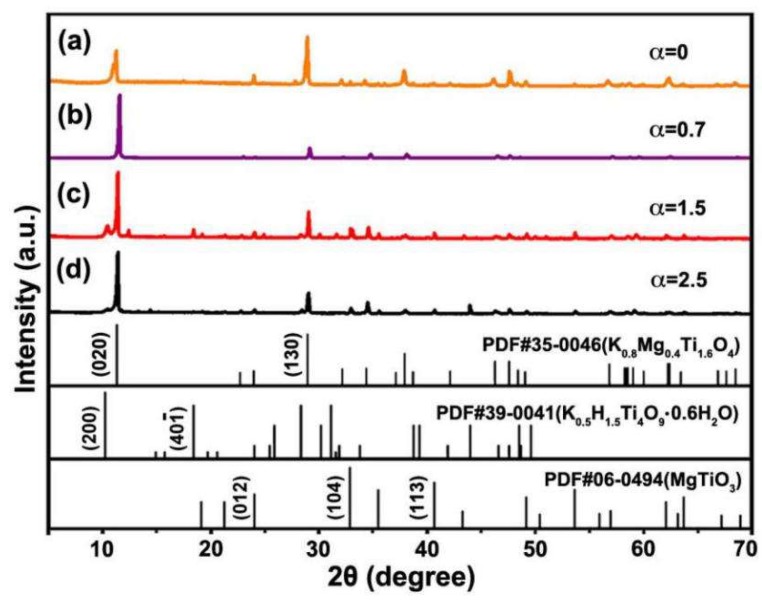
X-ray diffraction (XRD) patterns of three differently shaped KMTO products prepared at 1050 °C for 4 h with different molar ratios of the molten salt to the raw materials: (**a**) Without molten salt; (**b**) platelets, α = 0.7, β = 0.75; (**c**) boards, α = 1.5, β = 0.75; and (**d**) bars α = 2.5, β = 0.75.

**Figure 3 materials-12-01577-f003:**
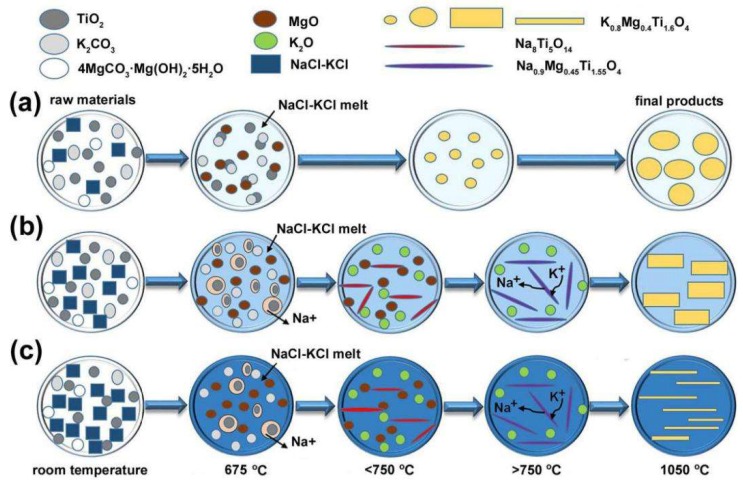
Schematic diagram of the synthesis mechanism of KMTO with different morphologies. (**a**) KMTO platelets, (**b**) KMTO boards, and (**c**) KMTO bars.

**Figure 4 materials-12-01577-f004:**
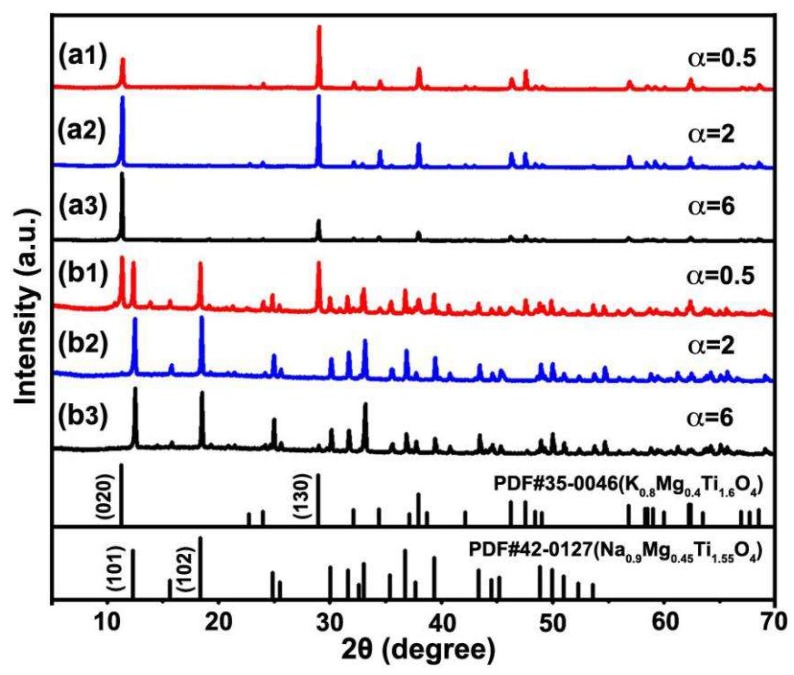
XRD patterns of samples prepared at 1050 °C for 4 h while using (**a1**–**a3**) KCl (β = 1) or (**b1**–**b3**) NaCl (β = 0) alone as the molten salt. (**a1**,**b1**): α = 0.5, (**a2**,**b2**): α = 2, and (**a3**,**b3**): α = 6.

**Figure 5 materials-12-01577-f005:**
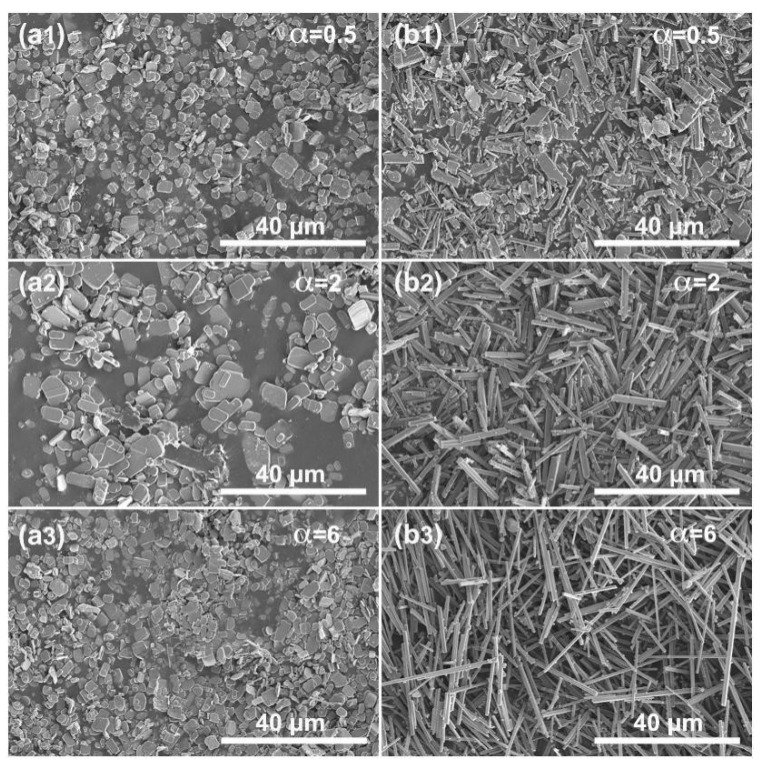
SEM images of the samples prepared at 1050 °C for 4 h while using (**a1**–**a3**) KCl (β = 1) or (**b1**–**b3**) NaCl (β = 0) alone as the molten salt. (**a1**,**b1**): α = 0.5, (**a2**,**b2**): α = 2, and (**a3**,**b3**): α = 6.

**Figure 6 materials-12-01577-f006:**
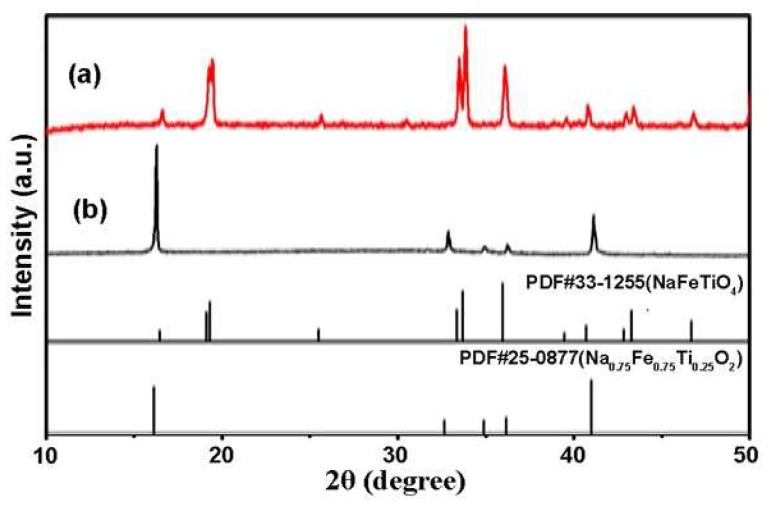
XRD patterns of (**a**) NaFeTiO_4_ needles (formed at α = 4, β = 0, T = 900 °C, and t = 4 h), and (**b**) Na_0.75_Fe_0.75_Ti_0.25_O_2_ platelets (formed at α = 4, β = 0, T=1000 °C, and t = 4 h).

**Figure 7 materials-12-01577-f007:**
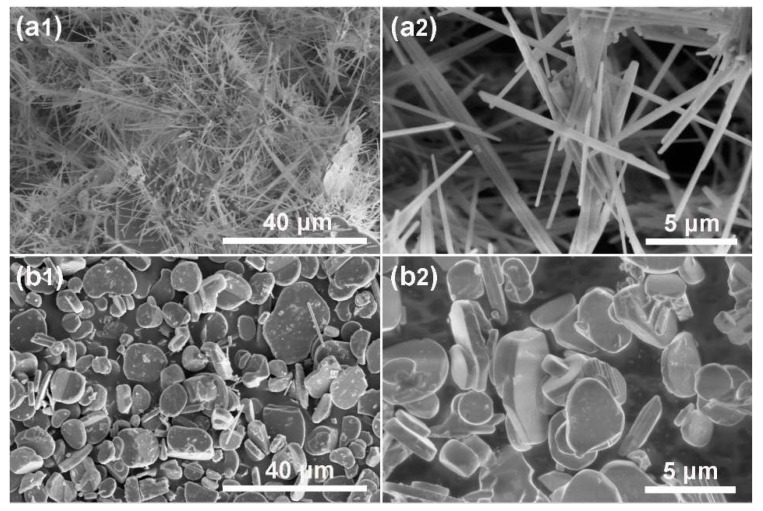
SEM images of (**a1**,**a2**) NaFeTiO_4_ needles (formed at α = 4, β = 0, T = 900 °C, and t = 4 h), and (**b1**,**b2**) Na_0.75_Fe_0.75_Ti_0.25_O_2_ platelets (formed at α = 4, β = 0, T=1000 °C, and t = 4 h).

**Figure 8 materials-12-01577-f008:**
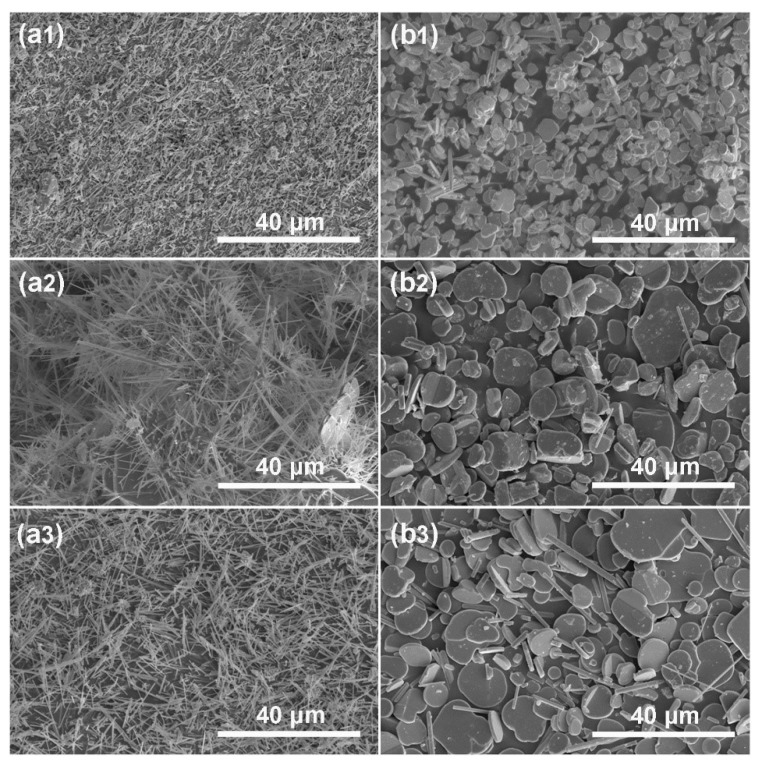
SEM images of (**a1–a3**) NaFeTiO_4_ needles and (**b1–b3**) Na_0.75_Fe_0.75_Ti_0.25_O_2_ platelets while using NaCl alone as the molten salt. (**a1**,**b1**) α = 2, (**a2**,**b2**) α = 4, and (**a3**,**b3**) α = 6; β = 0.

**Figure 9 materials-12-01577-f009:**
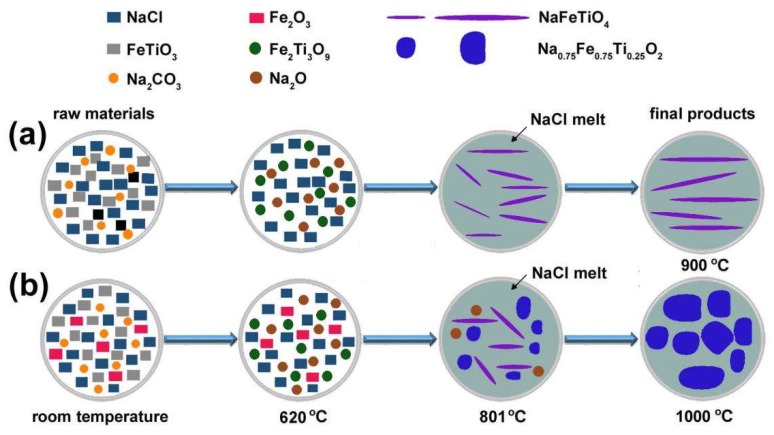
Schematic diagram of the synthesis mechanism of NFTO with different morphologies. (**a**) NaFeTiO_3_ needles, and (**b**) Na_0.75_Fe_0.75_Ti_0.25_O_2_ platelets.

**Figure 10 materials-12-01577-f010:**
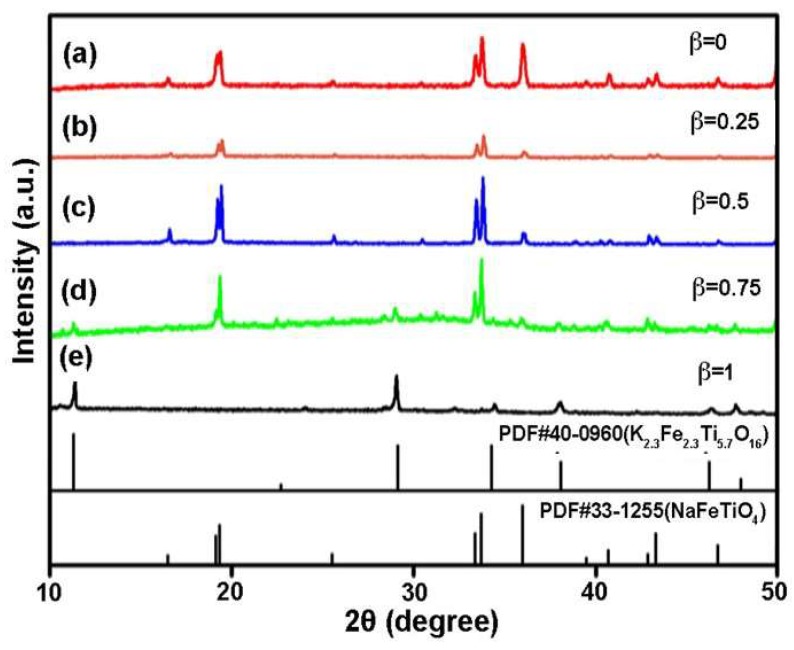
XRD patterns of samples obtained while using KCl–NaCl as molten salt at the condition of Na:Fe:Ti = 1.3:1:1, α = 4, T = 900 °C, and t = 4 h. (**a**) β = 0, (**b**) β = 0.25, (**c**) β = 0.5, (**d**) β = 0.75, and (**e**) β = 1.

**Figure 11 materials-12-01577-f011:**
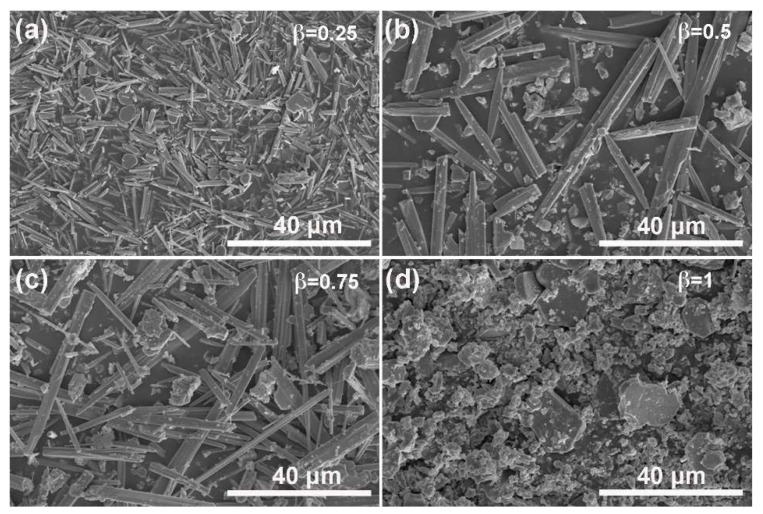
SEM images of samples obtained while using KCl–NaCl as molten salt at the condition of Na:Fe:Ti = 1.3:1:1, α = 4, T = 900 °C, and t = 4 h. (**a**) β = 0.25, (**b**) β = 0.5, (**c**) β = 0.75, and (**d**) β = 1.

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
