# Peer review of "Microstructure and Morphology Control of Potassium Magnesium Titanates and Sodium Iron Titanates by Molten Salt Synthesis"

_materials, 2019, doi:10.3390/ma12101577_

Reviewer 1 Report

Comments:

1.      It is necessary to indicate the material of crucible for preparation of magnesium titanates and iron titanates.

2.      What is the ratio of KCl to NaCl in the mixture of KCl-NaCl (line 100)?

3.      Please, described the procedure of reaction optimization (temperature and duration).

4.      The authors observed at the certain conditions the transfer of NMTO to KMTO (lines 158-160). At the same time the opposite process of transfer KMTO to NMTO was described too (lines 175-178). The thermodynamics of the exchange of sodium cations for potassium and vice versa should be calculated.

5.      Why the Na0.75Fe0.75Ti0.25O2 platelets and a small amount of NaFeTiO4 rods are both present in products at 900 °C, but at 1000 °C only the phase Na0.75Fe0.75Ti0.25O2 is exists (lines 220-223). The stability and formation of phases should be explained from the thermodynamic point of view.

Author Response

Thank you for your review, concerning the revision of our manuscript entitled “Microstructure and Morphology Control of Potassium Magnesium Titanates and Sodium Iron Titanates by Molten Salt Synthesis", to be considered for publication as an article in the Materials. We have made substantial changes to the manuscript based on your  suggestions and comments.

Enclosures, please find “Response to Reviewer 1 Comments”. The changes to the manuscript itself have been highlighted in red.

We appreciate your insightful and constructive suggestions and comments to help us improve the manuscript. We sincerely hope that our revisions have addressed you satisfactory.

Please let me know if any additional revision is needed.

Reviewer 2 Report

The compounds investigated in this paper showed nice quality and are very interesting from the points of view of the materials used in Na-ion and K-ion batteries. There are many compounds which researcher want to synthesize in this field. Therefore this work of synthesis methods are also very interesting. These investigations of synthesis for these compounds seem to be tough and good elaborated works.

L98-99: “the molar ratio of molten salt to raw materials”. Dose “the mole of raw materials” means sum of moles of raw materials? I am afraid that the reader difficult to reproduce your experiments. Is it preferable as “the molar ratio of molten salt to objective compound”?

Author Response

Thank you for your review, concerning the revision of our manuscript entitled “Microstructure and Morphology Control of Potassium Magnesium Titanates and Sodium Iron Titanates by Molten Salt Synthesis”, to be considered for publication as an article in the Materials. We have made substantial changes to the manuscript based on your  suggestions and comments.

Enclosures, please find “Response to Reviewer 2 Comments”. The changes to the manuscript itself have been highlighted in red.

We appreciate your insightful and constructive suggestions and comments to help us improve the manuscript. We sincerely hope that our revisions have addressed you satisfactory.

Please let me know if any additional revision is needed.

Round  2

Reviewer 1 Report

Comments:

1.      Line 105, should be KCl instead of KC

2.      The equilibrium diagram KCl-NaCl has not a eutectic point and in the text (line 162) should be written melting point instead of eutectic point.

Author Response

Point 1: Line 105, should be KCl instead of KC.

Response 1: Thank you for your careful check. “KC” has been changed to “KCl-NaCl” in page 3, line 105.

Point 2: The equilibrium diagram KCl-NaCl has not a eutectic point and in the text (line 162) should be written melting point instead of eutectic point.

Response 2: Thank you very much for pointing it out. It should be the melting point. We have revised it in Page 5, line 162.